# OpenReview forum: "Contrastive losses as generalized models of global epistasis"
_ICLR.cc/2024/Conference — Submitted to ICLR 2024_

### Official Review · Reviewer_68Sm · 2023-10-18

**Soundness:** 2 fair
**Presentation:** 3 good
**Contribution:** 2 fair
**Rating:** 6
**Confidence:** 4

**Summary:**

A global epistasis model for protein fitness assumes that the observed experimental fitness is g(f(x)), where x is the protein sequence, f is a simple (perhaps additive) function of the sequence, and g is a scalar -> scalar nonlinear function that reflects the non-linearity of the measurement process. The goal is to identify the latent function f(). They explore the use of learning-to-rank losses when fitting this model, as these losses are invariant to any monotone g(). They draw on some results from the compressed sensing literature, some experiments on synthetic data, and some experiments on a standard protein fitness prediction benchmark to argue that fitting models with a loss based on the Bradley-Terry ranking model is better than using mean squared error.

**Strengths:**

I liked section 2.1. I think that the 'epistatic domain' is a nice formalism for reasoning about a particular kind of sparsity of the fitness landscape, and I'll use this in the future.
I appreciate that the paper calls more attention to global epistasis. It's an elegant idea that is often overlooked.

**Weaknesses:**

All of the experiments evaluate fitness prediction in terms of spearman correlation. It's not surprising that training using a BT loss improves spearman correlation vs. MSE, since the BT loss can be seen as a differentiable relaxation of the negative spearman correlation. Therefore, I didn't find this result particularly exciting. It is a common observation in machine learning that a particular downstream metric can be improved by using a loss function that approximates it, and the learning-to-rank framework described in this paper is well established.

Similarly, I don't agree with the assertion that spearman correlation is a good surrogate metric for a model's usefulness for protein engineering. Suppose that fitness values appear in the range [0, 10] and that the best fitness seen so far in a protein engineering project is 6. Since we want to use this model to design new proteins, we don't care if it predicts 3 for a protein that actually has fitness 2. On the other hand, predicting 5.5 for a protein that has true fitness 6.5 could lead to a missed opportunity for ML-guided design. Good eval metrics should think asymmetrically about precision vs. recall of finding proteins with good fitness, etc. Finally, spearman correlation is confusing because it's unclear what the optimal value is. Given the noise level of the assay, I know the optimal MSE. What's the optimal spearman correlation, though?

The paper's goal is to fit models on experimental data to estimate a latent fitness function. However, the exposition assumes that these observations are noiseless, which is quite unrealistic. Further, in my experience noise is often heteroskedastic, where the noise level can be assumed to depend on g(f()). For example, the read-out from many assays is baesed on counts from DNA sequencing, and these are subject to Poisson noise. The BT loss strikes me as being not particularly robust to noise, since it could easily flip the binary fitness(A) > fitness(B) label if the noise is large relative to the difference between the fitness of A and B.

Section 3.2  on the 'fitness-epistasis uncertainty principle' is grounded in some results from the compressed sensing literature. It's interesting background on CS, and I enjoyed reading it, but I found the conclusions from the section too informal to be useful.

I found that the paper wasn't sufficiently grounded in standard statistical terminology about model estimation. Can you please discuss the identifiability of the global epistasis model? Does the change of loss function change identifiability? Also, you claim that the BT loss will result in better sample complexity. Maximum likelihood estimation has optimal sample complexity for parameter identification. Can you discuss the suitability of the BT loss in terms of whether the BT model reflects the observation process for the data? Is the goal to identify the latent function f()? What does that mean when f() is only identifiable up to a monotonic transformation (since the inverse transformation could be absorbed into g)?

In my prior reading on global epistasis, I hadn't seen the assumption that g() was monotonic. In many situtations, this isn't the case. For example, for many genes in the body it is hazardous if too much or too little of it is expressed, so g() is an upside-down U. In my experience with enzyme engineering, if an enzyme is too active it may kill the host cell used to express it. It seems that your framework does not generalize to this case.

**Questions:**

Can you please respond to my 'weaknesses' section?

How would your theoretical results or results on N-K landscapes change if the observed fitness values had noise added?


===========Update after extensive discussion with the authors=============

Thank you for such a thorough discussion in the author's response period. I appreciate the work you put in and have raised my review to a weak accept. Note that the paper is borderline, and this certainly may not result in acceptance to the conference.

Also, I apologize for not posting earlier, the discussion among reviewers was cut short because some administrative delays. The reviewers are currently discussing the paper.

Here is a summary of my assessment. Pros:1) I like that the paper brings attention to thinking about the targets of regression problems in the 'epistatic domain'. This can help illuminate the strengths and weaknesses of diffe rent modeling technique and could be helpful in other application domains. 2) I appreciate that the model gets improved performance on a couple of standard benchmarks. 3) The proposed modeling idea (using a learning-to-rank loss) is well motivated and relatively easy to implement.

Cons:

The methodological novelty of the paper is quite limited for the ICLR audience. Using learning-to-rank losses is standard in other application domains.
The synthetic data experiments focus too much on the setting where there is a lot of training data, which is not possible in many real-world situtations. I would have appreciated some empirical analysis of the bias-variance tradeoff of your technique. How does the performance scale with the amount of train data? Note that I did not ask for this analysis in my initial review, however, so I'm not taking this into consideration much for my decision.

---

> ### Author Response · Authors · 2023-11-16
> **Author response to Reviewer 68Sm (part 1)**
>
> We thank the reviewer for the thoughtful comments. We are eager to work with the reviewer to improve the paper and are hopeful that our point-by-point response below will help alleviate their most pressing concerns.
>
> > All of the experiments evaluate fitness prediction in terms of spearman correlation. It's not surprising that training using a BT loss improves spearman correlation vs. MSE, since the BT loss can be seen as a differentiable relaxation of the negative spearman correlation. Therefore, I didn't find this result particularly exciting. It is a common observation in machine learning that a particular downstream metric can be improved by using a loss function that approximates it, and the learning-to-rank framework described in this paper is well established.
>
> We agree with the reviewer that the improved performance of the BT loss over the MSE loss in terms of Spearman correlation is intuitive. However, the BT loss has not been used for fitness prediction previously and we were able to achieve improved performance on a metric that is common in the field (Spearman correlation). This is still an important contribution, even if the result is intuitive in retrospect.
>
> Further, the improved performance in benchmark tasks of models trained with the BT loss is not the only contribution of the paper. We have demonstrated that the BT loss is an effective technique for modeling global epistasis that allows one to recover a latent fitness function. Recovering such a latent fitness function is a problem that has been approached by a number of previous papers (Sailer & Harms, 2017 and Otwinowski et al., 2018 being two prominent examples), and we have shown that our method can do so while making fewer assumptions about the form of the global epistasis nonlinearity than these previous works (which are assumed to be power transforms or 3-splines in the aforementioned papers). The task of recovering a latent fitness function is important because it removes the spurious, or “phantom”, epistatic interactions that appear in a fitness function that has been affected by global epistasis, allowing for proper interpretation of the primary interactions that drive fitness. The latent fitness function is often interesting biologically as it can be a better measure of biophysical properties when the global non-linearity is caused by an artifact of the measurement process. Effectively solving this problem is a contribution of our paper, outside of the performance on benchmark tasks. We will clarify this in the introduction and discussion of our paper.
>
> Please see the following comments for the rest of our response to the reviewer.

---

> ### Author Response · Authors · 2023-11-16
> **Author response to Reviewer 68Sm (part 2)**
>
> > Similarly, I don't agree with the assertion that spearman correlation is a good surrogate metric for a model's usefulness for protein engineering. Suppose that fitness values appear in the range [0, 10] and that the best fitness seen so far in a protein engineering project is 6. Since we want to use this model to design new proteins, we don't care if it predicts 3 for a protein that actually has fitness 2. On the other hand, predicting 5.5 for a protein that has true fitness 6.5 could lead to a missed opportunity for ML-guided design. Good eval metrics should think asymmetrically about precision vs. recall of finding proteins with good fitness, etc.
>
> We have used test set Spearman correlation as our primary metric for assessing model performance because it is commonly used in benchmarking fitness prediction methods (see the [FLIP](https://benchmark.protein.properties) and [ProteinGym](https://www.proteingym.org/home) benchmarks for two prominent examples that employ Spearman correlation as the primary metric). We will clarify this in the paper, as it is not explicitly stated as our motivation for using Spearman correlation.
>
> However, we agree with the reviewer that the primary concern for protein engineering is tail performance rather than bulk correlation. In order to strengthen the results of the paper, we have now done an initial comparison of the tail performance of models trained with the BT and MSE losses on the GB1 predictions tasks in the FLIP benchmark. We follow [Gelman et al., 2021](https://www.pnas.org/doi/10.1073/pnas.2104878118) and consider the recall of the models in classifying the top sequences in the test set. In particular, we calculate the top 10% recall as the fraction of the sequences in the true top 10% of sequences in the test set that are in the top 10% of model predictions on the test set. For all splits except the Sampled split, models trained with the BT loss achieve statistically significant (p < 0.05) improvements over models trained with the MSE loss. In the case of the Sampled split, the MSE loss performs slightly better, but the difference is not statistically significant. The results of this analysis are summarized in the table below, where entries in the table correspond to the mean +/- std.dev. of top 10% recall over 10 random restarts of the models:
>
> | Split   | Bradley-Terry | MSE |
> | ----------- | ----------- | -----------|
> | Low-vs-High | 0.442 +/- 0.024  | 0.381 +/- 0.028 |
> | 1-vs-Rest | 0.138 +/- 0.051| 0.097 +/- 0.030 |
> | 2-vs-Rest | 0.282 +/- 0.008| 0.250 +/- 0.030 |
> | 3-vs-Rest | 0.664 +/- 0.014| 0.539 +/- 0.084 |
> | Sampled | 0.815 +/- 0.010| 0.823 +/- 0.010 |
>
> We will add these results, as well as corresponding results for the AAV and Meltome FLIP benchmarks to the Appendix and discuss the importance of metrics such as the top 10% recall for protein engineering in the main text.
>
> > Finally, spearman correlation is confusing because it's unclear what the optimal value is. Given the noise level of the assay, I know the optimal MSE. What's the optimal spearman correlation, though?
>
> In the paper’s noiseless simulations, the optimal spearman is 1. Given a noise level of an assay, it is possible to find the optimal spearman by parametric resampling of the noise on the test set. In realistic fitness prediction scenarios, the correct noise model is unknown and complex so one cannot know either the optimal Spearman nor MSE. The reviewer’s concern of course applies to all papers that use Spearman correlation as a metric, and it is outside the scope of our paper to resolve this broader concern.
>
> Please see the following comments for the rest of our response to the reviewer.

---

> > ### Author Response · Authors · 2023-11-16
> > **Author response to Reviewer 68Sm (part 3)**
> >
> > > Section 3.2 on the 'fitness-epistasis uncertainty principle' is grounded in some results from the compressed sensing literature. It's interesting background on CS, and I enjoyed reading it, but I found the conclusions from the section too informal to be useful.
> >
> > This section is meant to provide intuition and hypotheses for two observations. First, it has been observed in a number of papers that global epistasis results in the appearance of spurious or “phantom” epistatic terms (see, e.g., Sailer & Harms, 2017 and Baez-Centurion et al., 2019 for two examples). The fitness-epistasis uncertainty principle provides a mathematical explanation for this phenomenon, by demonstrating that the density of the epistatic representation must increase as the fitness domain becomes more concentrated (which is the effect of many common nonlinearities, as we show in our simulations). Second, we observe in our simulations that fitness functions affected by global epistasis require more data in order to effectively estimate with MSE loss. The uncertainty principle and the connection to the CS literature provide a potential hypothesis to explain this phenomenon; we have been careful to specify that this is only a hypothesis rather than a rigorous explanation. We believe that providing reasonable hypotheses to explain our observations is a valuable first step in moving towards a better understanding of these phenomena, even if these hypotheses are not proven in this paper.
> >
> > > I found that the paper wasn't sufficiently grounded in standard statistical terminology about model estimation. Can you please discuss the identifiability of the global epistasis model? Does the change of loss function change identifiability? Also, you claim that the BT loss will result in better sample complexity. Maximum likelihood estimation has optimal sample complexity for parameter identification. Can you discuss the suitability of the BT loss in terms of whether the BT model reflects the observation process for the data? Is the goal to identify the latent function f()? What does that mean when f() is only identifiable up to a monotonic transformation (since the inverse transformation could be absorbed into g)?
> >
> > We understand the desire for more standard statistical terminology, as we wrote this paper primarily from the perspective of signal processing. Our focus was on the practical advantages in performance and the effective identifiability of the latent fitness. BT loss is not invariant to monotonic transformations of f(x), (it is invariant to monotonic transformations of the data y). Clearly our results in Fig 1 and Fig 4 show effective identifiability of the latent fitness function with BT loss, whereas MSE loss has no such capability. Identifiability of the parameters of f(x) is impractical and unnecessary for neural networks, so we do not attempt this. We hypothesize that the BT loss is closer to the data generating process in the sense that some real global epistasis exists in these artificial and real fitness landscapes, and this results in better sample complexity.  We would welcome a formal treatment of the global epistasis problem; however, this is outside the scope of our paper, which we see as a complement to the existing literature on modeling global epistasis.
> >
> > > In my prior reading on global epistasis, I hadn't seen the assumption that g() was monotonic. In many situtations, this isn't the case. For example, for many genes in the body it is hazardous if too much or too little of it is expressed, so g() is an upside-down U. In my experience with enzyme engineering, if an enzyme is too active it may kill the host cell used to express it. It seems that your framework does not generalize to this case.
> >
> > The assumption of monotonicity in the global epistasis nonlinearity is very common among work that proposes general purpose methods for modeling global epistasis (see Sailer & Harms, 2017, Otwinowski et al., 2018 and Tareen et al., 2022 for three prominent examples). While not physically grounded, the monotonicity assumption is meant to enforce a constraint on the nonlinearity that allows the latent fitness function to be interpretable.  If the nonlinearity is allowed to be arbitrarily complex, then the latent fitness function could simply be random noise, with the nonlinearity containing all of the information about the fitness function.  A possible alternative would be to consider a smoothness constraint, but we are not aware of any work that has considered this and it is outside the scope of our paper to develop such a method. Instead, we consider our work as a complement to the existing literature on modeling global epistasis in which the monotonicity assumption is made.
> >
> > In cases such as that mentioned by the reviewer where one has*a priori* knowledge about the shape of the nonlinearity, then a bespoke method could easily be implemented to model this nonlinearity (e.g. by enforcing that the nonlinearity is an inverse parabola).

---

> > > ### Comment · Reviewer_68Sm · 2023-11-20
> > >
> > > Thanks for the detailed response.
> > >
> > > I think we all agree that the use of spearman correlation in the literature is fraught, and that it's worth using it for the sake of comparison to prior work, as long as we consider other evaluation metrics too.
> > >
> > > In my initial review I asked for some clarification about how your proposed results on recovery of the latent fitness function extend to a practical setting where there is noise in the assay. It would be easy to add noise in your simulation environment. How does that change things? In particular, noise occurs in two places. The latent fitness function is noisy (e.g., cells can be at different points in their life cycle and this affects all sorts of genotype-fitness relationships) and the experimental readout is noisy (due to NGS, masspec, etc).
> > >
> > > Also, I don't see a satisfactory definition of what it means to 'recover' the latent fitness function. Can you please formally define this?

---

> > > > ### Author Response · Authors · 2023-11-21
> > > > **Response to follow-up comments**
> > > >
> > > > Thank you for the follow-up comments.
> > > >
> > > > > In my initial review I asked for some clarification about how your proposed results on recovery of the latent fitness function extend to a practical setting where there is noise in the assay. It would be easy to add noise in your simulation environment. How does that change things?
> > > >
> > > > We have now completed simulations identical to those whose results are shown in Figure 2c but with homoskedastic noise Gaussian noise added to the observed fitness function. We tested two noise settings, with standard deviations of 0.025 and 0.1. These were selected such that the ranking of sequences would be substantially affected; the Spearmans between the latent and noisy fitness function were about 0.6 and 0.4 in the stddev=0.025 and 0.1 cases, respectively. In both of these cases, the Bradley-Terry loss maintains its advantage over the MSE loss in terms of Spearman correlation at all training set sizes. For example, at a training set size of 200, the average Spearman over simulation replicates are:
> > > >
> > > > | Noise std. dev.   | Bradley-Terry | MSE |
> > > > | ----------- | ----------- | -----------|
> > > > |0.025 | 0.513  |0.448 |
> > > > | 0.1 |0.296 | 0.278 |
> > > >
> > > > In the revised manuscript, the complete curves analogous to those shown in Figure 2c for these simulations will be included in the Appendix and referenced in the Discussion section.
> > > >
> > > > > In particular, noise occurs in two places. The latent fitness function is noisy (e.g., cells can be at different points in their life cycle and this affects all sorts of genotype-fitness relationships) and the experimental readout is noisy (due to NGS, masspec, etc).
> > > >
> > > > The reviewer's point about the multiple layers of noise is interesting and well-taken. Unfortunately a simulation of this situation would likely contain too many parameters to yield useful conclusions, so we rely on the results on the FLIP benchmark to report on the BT losses's performance in realistic scenarios.
> > > >
> > > > > Also, I don't see a satisfactory definition of what it means to 'recover' the latent fitness function. Can you please formally define this?
> > > >
> > > > The implicit definition used in Figure 1 is that the estimated fitness function is equal to the latent fitness function up to an affine transformation (hence our use of an R^2 metric in Figure 1c). In our view, this is appropriate definition because the optimal solution maintains the relative importance of interactions in the epistatic domain. We will make this clear in the revised manuscript.

---

### Official Review · Reviewer_7udB · 2023-11-01

**Soundness:** 3 good
**Presentation:** 3 good
**Contribution:** 3 good
**Rating:** 6
**Confidence:** 3

**Summary:**

In this study, optimization of Bradley-Terry (BT) contrastive loss is proposed to recover the latent fitness function corrupted by the effect of global epistasis. The proposed approach relies on the monotonic property of the global epistatic effect and does not require any assumption on the exact model of global epistasis. With simulated data, it is shown that BT loss achieves better estimation of fitness function than MSE loss, and it is more robust to the extent of corruption, measured by the entropy of epistatic representation, caused by the global epistasis. The advantage of using BT loss is also demonstrated for the protein fitness prediction on different splits of GB1, AAV, and thermostability datasets designed by the FLIP benchmark.

**Strengths:**

1)	Demonstrating the benefits of a ranking based loss for fitness prediction
2)	Nice experiments were designed to show the advantage of the contrastive loss in recovering the latent fitness model corrupted by global epistasis.

**Weaknesses:**

See questions below

**Questions:**

1)	What statistical test was used to measure the significance of improvements in Figure 3? I am asking this because in some splits the performance of BT loss is almost the same as MSE loss, but the reported p-value is significant (examples: samples and 7-vs-rest in AAV).
2)	In the splits where BT loss provides better performance (Figure 3), it is hypothesized that **it could be partially due to the corruption of fitness function with global epistasis**. I am curious to know how this can be proved.
3)	In Figure 1, are the coefficients in the epistatic domain normalized? The scale of $\hat{f}$ does not match f, which is expected.
4)	Have you also tested BT loss on fitness prediction for other datasets such as the ones compiled by the DeepSequence paper (https://www.nature.com/articles/s41592-018-0138-4)? I understand the use of FLIP datasets for the task of benchmark, however I am not sure how challenging FLIP splits are compared to other datasets out there.
5)	Where do you expect ranking-based losses not to perform as good as MSE losses for protein fitness prediction?

---

> ### Author Response · Authors · 2023-11-15
> **Response to Reviewer 7udB (part 1)**
>
> We thank the reviewer for the helpful feedback. We are hopeful that we can resolve each of the reviewer’s concerns in the specific responses below:
>
> > 1. What statistical test was used to measure the significance of improvements in Figure 3? I am asking this because in some splits the performance of BT loss is almost the same as MSE loss, but the reported p-value is significant (examples: samples and 7-vs-rest in AAV).
>
> We used a standard independent t-test with Bonferroni correction for multiple comparisons. Despite the bars having the visually similar heights, the differences between the methods are significant because the error bars over the random restarts for many of the splits are very small. For example, in the AAV “Sampled” split, the standard deviations of spearman correlations for the MSE and Bradley Terry losses are about 7e-4 and 4e-4, respectively. We will add a Table to the Appendix containing the numerical means and standard deviations for all of the bars shown in Figure 3, in order to clarify why the differences are significant.
>
> > 2. In the splits where BT loss provides better performance (Figure 3), it is hypothesized that it could be partially due to the corruption of fitness function with global epistasis. I am curious to know how this can be proved.
>
> We do not know of a straightforward path towards proving or disproving this hypothesis. The hypothesis was based on the analogy between the improved performance of the Bradley-Terry loss over the MSE loss in the simulated case when global epistasis was present, and the improved performance of the Bradley-Terry loss over the MSE loss in the case of real data. There may be multiple reasons for the improved performance in real data, and we do not immediately know how we could prove that the improved performance is due to the presence of global epistasis. Therefore, we will remove this sentence from the text in order to avoid making an untestable hypothesis.
>
> > 3. In Figure 1, are the coefficients in the epistatic domain normalized? The scale of f_hat does not match f, which is expected.”
>
> Yes, the coefficients in the epistatic domain are normalized such that the sum of the squared magnitudes sum to one. We will correct the axis label and figure caption to clarify this.
>
> Please see the next comment for the rest of our response.

---

> > ### Author Response · Authors · 2023-11-15
> > **Response to Reviewer 7udB (part 2)**
> >
> > Our response is continued below.
> >
> > > 4. Have you also tested BT loss on fitness prediction for other datasets such as the ones compiled by the DeepSequence paper (https://www.nature.com/articles/s41592-018-0138-4)? I understand the use of FLIP datasets for the task of benchmark, however I am not sure how challenging FLIP splits are compared to other datasets out there.”
> >
> > The most relevant benchmark outside of FLIP is ProteinGym, which is similar to but larger than the benchmark introduced in the DeepSequence paper (see [Notin et al., 2022](https://proceedings.mlr.press/v162/notin22a.html) for a comparison between the two benchmarks (the two papers share authors)). ProteinGym was curated for the purpose of benchmarking zero-shot prediction from pre-trained Protein Language models and thus is less appropriate for our use case than FLIP, which was curated for benchmarking supervised modeling of fitness data. The most significant difference between the two benchmarks is that FLIP curates specific dataset splits that mimic common situations in protein engineering (e.g. the two-vs-many split in which one predicts the fitness of sequences with 3 or more mutations using data only from sequences 2 or fewer mutations), while ProteinGym only curates complete datasets which must be split by the user. Therefore, only standard uniform splits can be tested in ProteinGym without substantially more work to curate more complex splits.  **It is important to note that the results from the FLIP benchmark paper indicate that the curated splits are strictly more difficult for supervised modeling approaches than uniform splits**. Therefore, FLIP provides a more challenging benchmark for supervised approaches such as training with the Bradley-Terry loss than ProteinGym.
> >
> > Nonetheless, we have now downloaded the ProteinGym benchmark data and begun to compare models trained with the MSE and Bradley-Terry losses on this data. At this time, we have only been able to run this comparison on one of the ProteinGym datasets: CAPSD_AAV2S_Sinai_substitutions_2021. For this dataset, we created 10 uniform 80/20 train/test splits and trained CNNs on each of the training datasets, using either the MSE or Bradley-Terry loss and all of the same model and training parameters as is used in the FLIP benchmark. We then calculated the Spearman correlation between the test set labels and model predictions in each corresponding test set. In this case, the models trained with the Bradley-Terry loss achieved a small but statistically-significant performance gain over the MSE loss (Spearman=0.9089 +/- 0.0038 and Spearman=0.9048 +/- 0.0037 for the Bradley-Terry and MSE losses, respectively, with p=0.027). If the reviewer thinks it will strengthen the paper, we are happy to run this test for more of the ProteinGym datasets and add a table to the appendix displaying the results, though we will be unlikely to test all 94 datasets due to time and compute constraints.
> >
> > > 5. Where do you expect ranking-based losses not to perform as good as MSE losses for protein fitness prediction?”
> >
> > A scenario in which we can expect ranking losses to perform worse than MSE for fitness prediction tasks is when the ranking of sets of sequences is determined by noise rather than underlying fitness values. In this case, we can expect the ranking losses to overfit to the noise, while the MSE loss will predict similar output values for all sequences in the set. However, we have found the Bradley-Terry loss to be surprisingly robust to noise, even in artificially pathological scenarios. In particular, in Appendix D we show a simulation in which the observed fitness values for sequences are given by y_i = I(x_i) + epsilon, where I(x) is a 0/1 indicator function and epsilon is Gaussian noise. We can expect this situation to be pathological for ranking losses, because the observed ranking within a class is entirely driven by noise. Unsurprisingly, we find that a model fit with an MSE loss on this data outperforms a model fit with the Bradley-Terry loss at classifying a test set between the 0/1 classes (AUC=0.977 and AUC=0.932 for the MSE and Bradley-Terry losses, respectively). Notably, the Bradley-Terry loss still performs quite well in this task. This Appendix D was mistakenly not referenced in the main text, but we will add a reference to it in the Discussion section and highlight that this is a regime where we can expect the ranking losses to perform worse than the MSE loss.

---

### Official Review · Reviewer_uFyB · 2023-11-09

**Soundness:** 3 good
**Presentation:** 3 good
**Contribution:** 3 good
**Rating:** 6
**Confidence:** 2

**Summary:**

In this paper, the authors focus on the problem of inferring fitness functions from experimental data, a relevant problem for protein engineering. To this end, the authors propose utilizing contrastive losses, such as the Bradley-Terry loss, to extract the underlying latent function from a global epistasis model. Furthermore, the authors argue that the choice of a contrastive loss may have other advantages of Mean Squared Error, especially for estimating ranking functions. They evaluate their approach on the FLIP dataset.

**Strengths:**

1. Contrastive learning has demonstrated promise in the field of computer vision; this paper shows a novel application of this concept to the field of Biology.

2. The paper has a strong theoretical foundation and is well presented.

3. Quantitative results are convincing and promising.

**Weaknesses:**

1. The empirical evaluation has focused only on a single benchmark, FLIP. It would strengthen the paper if the approach was validated with even more datasets.

**Questions:**

1. Is there any way to extend the empirical evaluation beyond the FLIP benchmark?

---

> ### Author Response · Authors · 2023-11-15
> **Response to Reviewer uFyB**
>
> We thank the reviewer for reading the paper and providing helpful comments. Responses to specific comments below:
>
> > Is there any way to extend the empirical evaluation beyond the FLIP benchmark?
>
> ProteinGym (https://www.proteingym.org/home) is another benchmark besides FLIP that can be used to validate sequence-to-fitness prediction models for proteins. This benchmark contains 94 curated datasets that relate sequences to experimentally-determined fitness values for a wide variety of proteins. ProteinGym is larger than FLIP; however, FLIP has a number of advantages that make it more appropriate for our use case:
> 1. FLIP curates specific dataset splits that are relevant for protein engineering (e.g. the two-vs-many split in which one predicts the fitness of sequences with 3 or more mutations using data only from sequences 2 or fewer mutations). These splits can be far more challenging for supervised models than a standard uniform split. In contrast, ProteinGym only curates complete datasets which the user must split themselves (usually by a standard uniform split). Therefore, FLIP provides a more challenging and diverse set of prediction tasks than ProteinGym, despite being an overall smaller benchmark.
> 2. ProteinGym was curated primarily to benchmark zero-shot prediction from pre-trained Protein Language Models. Therefore, the benchmark results that are presented on the ProteinGym webpage are not directly comparable to results from training a supervised model on the data. In contrast, FLIP was curated specifically to benchmark supervised sequence-to-fitness models for proteins, and therefore our results are directly comparable to those presented in the FLIP paper.
>
> Despite these advantages of FLIP, we have now downloaded the ProteinGym benchmark data and begun to compare models trained with the MSE and Bradley-Terry losses on this data. At this time, we have only been able to run this comparison on one of the ProteinGym datasets: CAPSD_AAV2S_Sinai_substitutions_2021. For this dataset, we created 10 uniform 80/20 train/test splits and trained CNNs on each of the training datasets, using either the MSE or Bradley-Terry loss and all of the same model and training parameters as is used in the FLIP benchmark. We then calculated the Spearman correlation between the test set labels and model predictions in each corresponding test set. In this case, the models trained with the Bradley-Terry loss achieved a small but statistically-significant performance gain over the MSE loss (Spearman=0.9089 +/- 0.0038 and Spearman=0.9048 +/- 0.0037 for the Bradley-Terry and MSE losses, respectively, with p=0.027). If the reviewer thinks it will strengthen the paper, then we are happy to run this test for more of the ProteinGym datasets and add a table to the appendix displaying the results, though we will be unlikely to test all 94 datasets due to time and compute constraints.

---

### Official Review · Reviewer_6v5p · 2023-11-10

**Soundness:** 3 good
**Presentation:** 3 good
**Contribution:** 2 fair
**Rating:** 6
**Confidence:** 3

**Summary:**

This study explores the estimation of fitness functions in protein engineering, which are complex mappings from biological sequences to properties of interest. The authors focus on global epistasis models that use a sparse latent fitness function transformed by a monotonic nonlinearity to predict measurable fitness.

Contribution: In this supervised learning setting, the authors introduce a rank-based contrastive loss approach, and show that it yields better results than an MSE loss-based approach (especially for small datasets), using both simulations and a benchmark dataset.

**Strengths:**

- Innovative use of supervised contrastive learning for fitness prediction.
- Empirical validation of proposed methods using simulations.

**Weaknesses:**

- presentation could be improved
- More results are needed in order to delineate the regimes where the presented results hold true
- Absence of simple baselines for comparative analysis

**Questions:**

1) Clarity on "Corrupting Data": While recognising that the term "corrupting data" might be specific jargon within the authors' field, I find its usage potentially confusing. It typically suggests that data has been made less accurate. In contrast, from a machine learning standpoint, the issues you're addressing appear to be related to the complexity introduced by non-linear relationships, which need complex models that may overfit, particularly when models are trained to predict the exact observed values (y).

2) It would be highly beneficial to demonstrate the specific regimes in which your observations about the Bradley-Terry loss are valid. Specifically, it's important to determine whether the improvements attributed to the BT loss over the MSE loss are unique to the complex models with epistatic interactions, or if similar benefits could be observed with simpler models, such as a linear model subjected to the same monotonic warping (i.e., the nonlinearity introduced by global epistasis). To address this, I recommend varying the degree of interaction order in your simulations.

3) I would recommend including a comparison with a more straightforward baseline to further validate the proposed approach. Specifically, it would be informative to see how a simple quantile transformation of the outcome data (to uniform or Gaussian distributions) performs in conjunction with Mean Squared Error (MSE) loss. This could serve as a more direct way to deal with the non-linear transformations introduced by global epistasis and might provide a competitive baseline to the BT loss approach.

4) Minor: To avoid confusion with unsupervised contrastive learning methods, it would be beneficial to explicitly state that the contrastive learning approach employed is supervised. (e.g., you could say "supervised contrastive learning loss")

If the authors can address these concerns and provide clarifications, I would be open to revisiting and potentially improving my score.

------------- After rebuttal ------------------
The authors addressed most of my major concern. Based on this I updated my score.

---

> ### Author Response · Authors · 2023-11-15
> **Response to Reviewer 6v5p**
>
> We thank the reviewer for the helpful feedback. We are eager to provide new experiments and writing clarifications to respond to each of the concerns of the reviewer. We respond to each question in turn below:
> 1. **On “corruption” language**: it is helpful to know that this is confusing language. We had meant to use it to refer to the effect that global epistasis produces a dense epistatic representation (similar to the effect of noise), which results in higher sample complexity. The reviewer’s point is well taken that this is a distinct negative outcome from the effects of noise and therefore we should not use “corruption” to refer to global epistasis. We will update the manuscript to remove this language.
> 2. **On testing our methods with varying degrees of interaction orders**: We agree that testing our method for different degrees of interaction order is useful. We repeated the simulation described in Section 3.1 (“Recovery from complete data”) with K=1 (i.e. an additive latent fitness function) and K=3 (up to third-order interactions), in addition to the K=2 setting used in the main text; the results of these additional simulations are shown in Appendix B of the initial submission, which was mistakenly not referenced in the main text (we will add a reference in Section 3.1). These results show that the complete data recovery result is maintained in for all of these interaction order regimes. Further, we have now repeated the simulation results described in Section 3.3 (“Simulations with incomplete data”) for K=1 and K=3 and these results will be added to the Appendix. We have created analogous figures to Figure 2b and 2c using the results from these K=1 and K=3 simulations. Comparing these to Figure 2b and 2c (which are based on simulations using K=2) we find that:
>
>    * The results in Figure 2b are qualitatively similar to the analogous results for the K=1 and K=3 simulations. In particular, the MSE loss shows a clear drop in performance as the entropy of the observed fitness function increases, while the Bradley-Terry loss is more robust to this increase in entropy.
>
>    * The results in Figure 2c are qualitatively similar to the analogous results for the K=1 and K=3 simulations. In the K=1 case, the distinction between the MSE and Bradley-Terry losses is not as large as in the K=2 and K=3 cases, because both losses are able to achieve high performance at small training set sizes (i.e. spearman > 0.9 at 100 training samples). However, if we increase the intensity of the nonlinearity by using a setting of alpha=20 instead of alpha=10, then the distinction between the losses is similar to that observed in Figure 2c. This indicates that the Bradley-Terry loss can still be useful for modeling additive latent fitness functions under global epistasis, but the nonlinearity of global epistasis must be more extreme to observe large performance gains over the MSE loss. The results of these simulations for the alpha=20 setting will also be added to the Appendix.
> 3. **On comparing to a baseline method based on quantile transformations**: We thank the reviewer for suggesting this baseline method. We have tested this baseline method in a simulation analogous to that described in Section 3.1 (“Recovery from complete data”); in particular we tested the ability of quantile transformation to recover a latent fitness function from complete data that has been affected by global epistasis. We found that quantile transformations do perform quite well at recovering the latent fitness function (R^2 =0.951 and R^2=0.963 for quantile transformations to normal and uniform distributions, respectively), but not as well as the Bradley-Terry loss (R^2=0.999, as seen in Figure 1c in the main text). Further, we are committed to testing the baseline method in the simulations described in Section 3.3 (“Simulations with incomplete data”) in order to compare the baseline to the Bradley-Terry loss in a more realistic scenario.
> We note that the baseline method described by the reviewer makes stronger assumptions about the nature of the latent fitness function than the Bradley-Terry loss, and this may be the reason for the difference in performance in the complete data case. In particular, the baseline method assumes the latent fitness function follows either a uniform or normal distribution, while the Bradley-Terry loss makes no such assumption. Considering that these methods are similar in their implementation difficulty (both require fewer than 10 lines of code), the Bradley-Terry loss is preferable because it requires fewer assumptions and appears to provide better performance.
> 4. **On clarifying that we use supervised contrastive losses**: It is helpful to know that this is a point of confusion. We will add a sentence to the Methods that clarifies that we use supervised rather than unsupervised contrastive losses, and reinforce this point by referring to “supervised contrastive losses” wherever appropriate.

---

> ### Comment · Reviewer_6v5p · 2023-11-15
> **Remaining concerns**
>
> Thank you for your detailed responses and for the openness to address the provided feedback.
>
> 1) Thanks, glad you found this comment helpful. Minor, but will make your work more accessible and impactful.
>
> 2) Regarding the effectiveness of the Bradley-Terry loss across varying interaction orders (K=1, K=2, and K=3), I've noticed a potential disconnect between the presentation in your paper and the data provided. It appears that the benefits of the Bradley-Terry loss are not solely tied to the complexities of epistasis (K>1), but are also evident in simpler scenarios such as K=1, which representing warped latent additive models. This observation suggests that the advantages of the Bradley-Terry loss may extend beyond handling epistatic interactions to include any scenario with non-Gaussian outcomes. It's crucial to explicitly address this in your paper, as it could significantly broaden the applicability and understanding of the Bradley-Terry loss's effectiveness.
>
> 3) In your response, you mention testing the ability of quantile transformation to recover a latent fitness function from complete data affected by global epistasis. I think the baselines should be quantile transformation, then training with MSE loss. Is this what you have done?
> On the comments you made on the number of lines of code: while the Bradley-Terry loss may be concise in terms of code implementation, it is essential to consider the broader aspects of simplicity, particularly in terms of conceptual understanding and practical application. The contrastive nature of the Bradley-Terry loss, focusing on pairwise comparisons, represents a paradigm shift from more traditional regression-based approaches. Rank-transformation + train with MSE remains the simple baseline to beat here.
>
> 4) My comment refers more to the abstract and intro. This is a supervised ML setting. It is beneficial to make this clear early in the paper. Btw, this is very minor at this point.

---

> > ### Author Response · Authors · 2023-11-16
> > **Author response to remaining concerns of Reviewer 6v5p**
> >
> > Thank you for the follow up comments. We respond to the major points below:
> >
> > > 2. Regarding the effectiveness of the Bradley-Terry loss across varying interaction orders (K=1, K=2, and K=3), I've noticed a potential disconnect between the presentation in your paper and the data provided. It appears that the benefits of the Bradley-Terry loss are not solely tied to the complexities of epistasis (K>1), but are also evident in simpler scenarios such as K=1, which representing warped latent additive models. This observation suggests that the advantages of the Bradley-Terry loss may extend beyond handling epistatic interactions to include any scenario with non-Gaussian outcomes. It's crucial to explicitly address this in your paper, as it could significantly broaden the applicability and understanding of the Bradley-Terry loss's effectiveness.
> >
> > The reviewer is correct that the benefits of the Bradley-Terry loss extend to the case of additive models that have been affected by global epistasis. This is an important problem because even additive models affected by global epistasis contain spurious epistatic interactions in the observed data, so recovering the underlying additive fitness function enables proper interpretation of the nature of the interactions (or lack thereof) that drive fitness. We will make sure to clarify this in the text.
> >
> > Further, our noiseless simulations in Fig 1 and 2 demonstrate that the effects of global epistasis and the ability of the Bradley-Terry loss to mitigate these effects is independent of noise, and therefore applies to both Gaussian and non-Gaussian outcomes.
> >
> > > 3.  In your response, you mention testing the ability of quantile transformation to recover a latent fitness function from complete data affected by global epistasis. I think the baselines should be quantile transformation, then training with MSE loss. Is this what you have done? On the comments you made on the number of lines of code: while the Bradley-Terry loss may be concise in terms of code implementation, it is essential to consider the broader aspects of simplicity, particularly in terms of conceptual understanding and practical application. The contrastive nature of the Bradley-Terry loss, focusing on pairwise comparisons, represents a paradigm shift from more traditional regression-based approaches. Rank-transformation + train with MSE remains the simple baseline to beat here.
> >
> > In the case of the “complete data” simulations whose results are shown in Fig 1 and that we referenced in the previous section, training a model with MSE will result in a model that perfectly recapitulates the observed data (there is no test data so there can be no overfitting). Therefore, for this initial test of the baseline it would be redundant to fit a model with MSE after performing the quantile transform. However, for the follow up experiments on this baseline with incomplete data (analogous to those described in Section 3.3) that we will perform, we will follow the reviewer’s proposed method and fit a model to the transformed data with MSE.
> >
> > We appreciate the reviewer pointing out a flaw in our arguments about simplicity. Indeed, the Bradley-Terry loss represents a paradigm shift despite its simplicity of implementation. We are committed to testing this against the more traditional baseline proposed by the reviewer. However, we would like to point out that the proposed baseline is somewhat inappropriate for modeling global epistasis in that the nonlinearity assumed by the method is not adaptive. In particular, the quantile transformation is a deterministic preprocessing step that assumes a particular nonlinearity. This is contrast to both the Bradley-Terry loss and previous methods for modeling global epistasis (such as the methods in Sailer and Harms, 2017 and Otwinowski et al., 2018) where the nonlinearity is modeled jointly with the latent fitness function. This joint modeling is crucial because the form of global epistasis nonlinearities are not generally known *a priori*.
> >
> > > My comment refers more to the abstract and intro. This is a supervised ML setting. It is beneficial to make this clear early in the paper. Btw, this is very minor at this point.
> >
> > Thank you for this clarification; we will make sure to specify that we use supervised contrastive losses in the abstract and introduction.

---

### Author Response · Authors · 2023-11-21
**Summary of revisions**

We have uploaded a revised manuscript in response to the comments of the reviewers. The summary of changes are:
1. In response to reviewer 6v5p, we have added to Appendix D.1 results of simulations in which we have changed the degree of epistatic interactions present in the latent fitness function. These results are qualitatively similar to the results in Figure 2, and are referenced in the main text.
2. We have added results to Appendix B.2 showing the performance of the baseline method suggested by reviewer 6v5p on the task described in Section 3.1, and referenced these results in the main text. In this test, the baseline method does not recover the latent fitness function as well as the Bradley-Terry loss. This is likely because the baseline is not adaptive, but rather assumes a particular form of the global epistasis nonlinearity.
3. In response to reviewer 68Sm, we have added results to Appendix E.2 that compare the ability of models to classify the highest fitness sequences in the FLIP benchmark, and discussed the importance of these metrics for protein engineering in the main text. In 9/15 of the splits, the BT loss significantly outperforms the MSE on this task (in 2 splits the MSE is significantly better and in 4 splits there is not a significant difference)
4. In response to reviewer 68Sm, we have added results to the Appendix D.2 that add noise to the simulations described in Section 3.3. The results of the simulations are qualitatively similar to those shown in Figure 2c. We have also added a discussion of noise to the Discussion section.
5. In response to reviewer 7udB's concern about statistical significance in Figure 3, we have added these results in table form to Appendix E.1. In this view, the error bars in the results are more clear.
6. We have changed our writing in a number of points in response to the reviewers, including removing "corruption" language (in response to 6v5p), clarifying that we are using supervised contrastive loss functions (in response to 6v5p), removing an untestable hypothesis (in response to 7udB), defined our definition of recovery of a latent fitness function in Section 3.1 (in response to 68Sm), and added references to appendices that were not previously referenced.

We believe that these changes address the reviewer's concerns and we look forward to hearing their response.

---

### Meta-Review · Area_Chair_GhNG · 2023-12-31

**Metareview:**

The present paper addresses the question of how to best learn a general biological fitness function from incomplete data, and suggests the use of the Bradley-Terry model to focus on rankings or relative assessments, rather than MSE point estimates as is currently common in the literature. The paper provides empirical evidence using simulations of clean and incomplete data, as well as on the FLIP dataset.

Reviews for this paper were initially divided, with reviewers appreciating the focus on global epistasis, noting the promise of contrastive losses for biology, the clearly demonstrated benefits using simulations. At the same time, many weaknesses were noted, including questions about the validity of the strong focus on spearman's rank correlation as the main evaluation metric, the need for more formal grounding in the statistical learning literature, and the need to validate the usefulness of the proposed approach across additional data sets.

The final recommendation takes into account the current version of the manuscripts, rebuttal and discussion between the authors and reviewers, as well as additional reviewer-AC discussion. All reviewers appreciated the authors' efforts to clarify and address their concerns. At the same time, the AC notes the remaining concerns. Most importantly, the key value in this paper is in shedding light on realistic uses of ranking models in the specific application of recovering latent fitness functions from sparse data, and as such, the empirical results should be more comprehensive than in the present form of the manuscript.

**Justification For Why Not Higher Score:**

This is a paper focusing on empirical benefits of relative losses for recovering latent fitness functions from sparse observations. Given this focus, the set of empirical insights provided in the current manuscript is not comprehensive enough to meet acceptance criteria.

**Justification For Why Not Lower Score:**

N/A

---

### Decision · Program_Chairs · 2024-01-16

Reject